Pollen extracts and constituent sugars increase growth of a trypanosomatid parasite of bumble bees

Palmer-Young Evan C. ecp52@cornell.edu 1
Thursfield Lucy 2
1 Organismic and Evolutionary Biology, University of Massachusetts at Amherst , Amherst , MA , United States of America
2 Royal Botanic Gardens, Kew , Richmond , Surrey , United Kingdom
Clayton Christine
Electronic publication date: 2017 May 9
Publication date: 2017
Volume: 5
Electronic Location ID: e3297
Received 2017 Jan 13; Accepted 2017 Apr 11
Copyright: ©2017 Palmer-Young and Thursfield
Copyright year: 2017
Copyright holder: Palmer-Young and Thursfield
License: This is an open access article distributed under the terms of the Creative Commons Attribution License, which permits unrestricted use, distribution, reproduction and adaptation in any medium and for any purpose provided that it is properly attributed. For attribution, the original author(s), title, publication source (PeerJ) and either DOI or URL of the article must be cited.
License URL: https://creativecommons.org/licenses/by/4.0/

Keywords: Bombus, Plant secondary metabolites, Parasite, Pollinator decline, Crithidia, Nutrient limitation, Antitrypanosomal, Tritrophic interactions, HPLC, Proline

Funding: USDA NIFA Predoctoral Fellowship 2016-67011-24698 This research was funded by the United States Department of Agriculture (USDA: usda.gov) Agricultural and Food Research Initiative (AFRI) Food, Agriculture, Natural Resources and Human Sciences Education and Literacy Initiative (ELI) Predoctoral Fellowship Award Number: 2016-67011-24698 to ECPY; and by the Garden Club of America (www.gcamerica.org) (Centennial Pollinator Fellowship to ECPY). Any opinions, findings, and conclusions or recommendations expressed in this material are those of the authors and do not necessarily reflect the views of the funding agencies. The funders had no role in study design, data collection and analysis, decision to publish, or preparation of the manuscript.

==============================
Phytochemicals produced by plants, including at flowers, function in protection against plant diseases, and have a long history of use against trypanosomatid infection. Floral nectar and pollen, the sole food sources for many species of insect pollinators, contain phytochemicals that have been shown to reduce trypanosomatid infection in bumble and honey bees when fed as isolated compounds. Nectar and pollen, however, consist of phytochemical mixtures, which can have greater antimicrobial activity than do single compounds. This study tested the hypothesis that pollen extracts would inhibit parasite growth. Extracts of six different pollens were tested for direct inhibitory activity against cell cultures of the bumble bee trypanosomatid gut parasite Crithidia bombi. Surprisingly, pollen extracts increased parasite growth rather than inhibiting it. Pollen extracts contained high concentrations of sugars, mainly the monosaccharides glucose and fructose. Experimental manipulations of growth media showed that supplemental monosaccharides (glucose and fructose) increased maximum cell density, while a common floral phytochemical (caffeic acid) with inhibitory activity against other trypanosomatids had only weak inhibitory effects on Crithidia bombi. These results indicate that, although pollen is essential for bees and other pollinators, pollen may promote growth of intestinal parasites that are uninhibited by pollen phytochemicals and, as a result, can benefit from the nutrients that pollen provides.

Introduction

Plants provide nutrients that sustain the growth and reproduction of many animal species, but also contain antimicrobial phytochemicals that may counteract infection in plant-eating animals (De Roode et al., 2013). Insect pollinators such as bumble bees, whose diets consist exclusively of phytochemical-rich nectar and pollen, are important for agricultural production. However, like honey bees, bumble bees are threatened by parasite-related decline (Goulson et al., 2015). Because bumble bees have abundant natural access to phytochemicals, antimicrobials from flowers could provide a natural source of medicinal compounds that could counteract infection in pollinator populations.

Trypanosomatids are parasites that, in addition to afflicting over 12 million humans, infect and decrease fitness of many species of insects (Maslov et al., 2013). For example, the newly described Jaenimonas drosophilae increases mortality of pupae and decreases fecundity of adult Drosophila melanogaster (Hamilton et al., 2015). Trypanosomatid infection can also be pervasive in populations of wild and managed bees (Shykoff & Schmid-Hempel, 1991; Schwarz et al., 2015). With spread of parasites facilitated by use of shared flowers (Durrer & Schmid-Hempel, 1994; Graystock, Goulson & Hughes, 2015), infection in some areas may exceed 80% in bumble bees (Shykoff & Schmid-Hempel, 1991; Gillespie, 2010; Popp, Erler & Lattorff, 2012) and 90% in honey bees (Runckel et al., 2011). Correlative evidence implicates trypanosomatid infection as a factor in honey bee colony loss. In Belgian honey bees, infection with Lotmaria passim (formerly named and reported as Crithidia mellificae (Schwarz et al., 2015)) was correlated with colony death (Ravoet et al., 2013). In the United States, Lotmaria passim infection intensity was over six-fold higher in hives that suffered from Colony Collapse Disorder than in hives that did not collapse (Cornman et al., 2012). In bumble bees, Crithidia bombi infection is similarly detrimental for individuals and colonies. Infection increased the rate of death in starved workers (Brown, Schmid-Hempel & Schmid-Hempel, 2003), provoked potentially costly immune responses (Sadd & Barribeau, 2013), altered foraging behavior and learning (Gegear, Otterstatter & Thomson, 2005), and decreased colony fitness (Shykoff & Schmid-Hempel, 1991).

Phytochemicals have well-known antimicrobial properties that inhibit infection not only in plants (Bennett & Wallsgrove, 1994; Huang et al., 2012), but also in animals that consume phytochemical-rich plant materials (Karban & English-Loeb, 1997; Singer, Mace & Bernays, 2009; De Roode et al., 2013). Plant-based therapeutics have a long history of traditional use against trypanosomatid infection, and recent studies have confirmed the inhibitory activity of both plant extracts and isolated phytochemicals against trypanosomatid cell cultures (Merschjohann & Steverding, 2006; Santoro et al., 2007a; Wink, 2012). Similarly, phytochemicals may have medicinal effects in bees infected with the trypanosomatid C. bombi (Manson, Otterstatter & Thomson, 2010; Baracchi, Brown & Chittka, 2015; Richardson et al., 2015; Biller et al., 2015), such that nectar and pollen of phytochemical-rich wildflowers and crops could provide medicinal resources for pollinators.

To date, all studies on the medicinal effects of phytochemicals on bees have tested single compounds. However, plants contain mixtures of phytochemicals that can have synergistic effects against both insects (Berenbaum & Neal, 1985; Berenbaum, Nitao & Zangerl, 1991) and microbes (Fewell & Roddick, 1993), including C. bombi (Palmer-Young et al., 2017). The defensive compounds in phytochemical-rich plants, such as milkweed (Danaus spp.) (Gowler et al., 2015), can also counteract pathogens of plant-eating insects and other animals (De Roode et al., 2013). In a mouse model of Plasmodium falciparum malaria, crude Artemisia annua plant extract had a stronger medicinal effect than did equivalent amounts of purified artemisinin (Elfawal et al., 2012). Artemisia spp. extracts can also have inhibitory effects against trypanosomatids, such as the C. bombi relative Leishmania major, where the effects of crude plant extracts and phytochemically complex essential oils can have greater inhibitory activity than do individual compounds (Efferth et al., 2011). Similarly, essential oil from Thymus vulgaris plants was a more powerful inhibitor of Trypanosoma cruzi growth than was the purified main constituent thymol (Santoro et al., 2007b). Together, these studies suggest that the phytochemical mixtures found in natural plant materials may be more effective inhibitors of parasites than are isolated chemicals.

Pollen and nectar consumed by bees contain diverse phytochemicals (Dobson & Bergstrom, 2000; Adler, 2001; Jakubska et al., 2005). For example, over 100 compounds were found in the nectar of a single orchid species (Jakubska et al., 2005), and bumble bees forage from a variety of floral species throughout the growing season (Goulson & Darvill, 2004; Heinrich, 2004; Vaudo et al., 2015). Pollen contains particularly high phytochemical concentrations that can exceed those in nectar by several orders of magnitude (Detzel & Wink, 1993; London-Shafir, Shafir & Eisikowitch, 2003; Palmer-Young et al., 2016). Hence, pollen could be expected to have particularly strong effects on parasites that are susceptible to inhibition by phytochemicals. However, studies in bees have found that, even though pollen consumption increased expression of immune genes (Brunner, Schmid-Hempel & Barribeau, 2014), dietary pollen increased C. bombi infection intensity (Logan, Ruiz-González & Brown, 2005; Conroy et al., 2016) in Bombus terrestris and B. impatiens. One hypothesis to explain the positive effects of pollen was that nutrients in pollen promote parasite growth. Pollen is rich in carbohydrates, proteins, and other nutrients that are essential for bee reproduction (Roulston & Cane, 2000), but these nutrients could also benefit parasites that can tolerate high phytochemical concentrations (Palmer-Young et al., 2016).

To test the alternative hypotheses that (a) pollen phytochemicals inhibit parasites and (b) pollen nutrients benefit parasites, we tested the direct effects of six pollens, a nectar phytochemical with demonstrated effects against trypanosomatids, and monosaccharides on C. bombi growth in cell culture.

Materials and Methods

Overview

Three experiments were conducted to elucidate the effects of pollen extracts on in vitro growth of C. bombi cell cultures. Experiments evaluated the effects of (1) extracts of single pollens, (2) extracts of mixed pollens, and (3) specific chemicals (sugar and the floral phytochemical caffeic acid) in order to better understand the mechanisms by which pollen extracts affected growth. In addition, the pollen extracts were chemically analyzed by HPLC to determine sugar content.

Parasite culturing

Crithidia bombi is a flagellated trypanosomatid parasite that infects bumble bees (Lipa & Triggiani, 1988; Schmid-Hempel & Tognazzo, 2010). Phylogenetic analyses (Schwarz et al., 2015) showed that C. bombi belongs to one of four clades in the subfamily Leishmaniinae, which is one of roughly 12 trypanosomatid subfamilies (Maslov et al., 2013). The clade of C. bombi also includes the honey bee parasites C mellificae and L. passim as well as Leptomonas spp.. Other clades within the Leishmaniinae include the Leishmania spp. human parasites and the mosquito-infecting model organism Crithidia fasciculata (Schwarz et al., 2015).

Crithidia bombi was isolated in 2013 by Ben Sadd from feces of wild Bombus impatiens near Normal, IL by flow cytometry (Salathé et al., 2012) and kept frozen at −80 °C until several weeks before the experiments. Thereafter, cells were grown at 27 °C in 50 cm2 tissue culture flasks and transferred to fresh media every 3–4 days. No special permits were required for the collection. Cultures are available upon request, provided appropriate documentation and permissions are supplied. The growth medium was composed of a tryptose-liver broth (containing 2.2 mM glucose) supplemented with B vitamins, haemin, and 10% heat-inactivated fetal bovine serum. In addition, the medium contained 10 mM fructose and 2.5 mM free proline (Salathé et al., 2012).

Pollen and phytochemical treatments

Six types of bee-collected pollen—buckwheat (Fagopyrum esculentum), lotus (Nelumbo nucifera), poppy (Papaver somniferum), rapeseed (Brassica napus), sunflower (Helianthus annuus), and tea (Camellia sinensis)—were obtained from Changge Huading Wax Industry (Henan, China) in 2015. The pollens were stored at −20 °C and sorted to remove heterogeneous granules. For extraction, 6 g of pollen was incubated for 24 h at room temperature in constant darkness with 20 mL of 50% v/v aqueous methanol in a 50 mL conical tube. The 50% methanol was used as a solvent due to its widespread application in phytochemical extraction of pollen (Serra Bonvehi, Soliva Torrentó & Centelles Lorente, 2001) and other plant tissues (Keinänen, Oldham & Baldwin, 2001). Samples were shaken at 180 rpm on a shaker table for the first 20 min of the extraction. After 24 h, tubes were centrifuged (30 min, 2,700 g) and the supernatant removed, sterile-filtered, aliquoted, and stored at −80 °C until use. The mixed-pollen extract consisted of equal volumes of buckwheat, rapeseed, and sunflower extracts, which were combined immediately before the experiment.

Caffeic acid was used as a representative phytochemical to evaluate possible negative and positive effects of pollen constituents on C. bombi. This hydroxycinnamic acid is likely to be widespread in bee diets, as it was the most widespread phytochemical in honey extracts, with occurrence in all 14 tested types of Turkish honey (Can et al., 2015); cinnamic acids and other phenolics are also abundant in pollen (Campos et al., 1997; Serra Bonvehi, Soliva Torrentó & Centelles Lorente, 2001; Almaraz-Abarca et al., 2004). Caffeic acid inhibited the trypanosomatids Leishmania donovani, Trypanosoma cruzi, and T. brucei (Tasdemir et al., 2006; Grecco et al., 2014), which suggested that caffeic acid could inhibit C. bombi as well. However, caffeic acid is also a powerful antioxidant, with ability to scavenge reactive oxygen species that exceeds that of ascorbic acid and is comparable to that of tocopherols (Chen & Ho, 1997; Almaraz-Abarca et al., 2004). Antioxidant activity of caffeic acid and other pollen components (Almaraz-Abarca et al., 2004) might protect Crithidia bombi from stress incurred during the experiment, such as shaking and handling, and in the wild, where parasites encounter temperature changes, osmotic shock, UV light, and pro-oxidant enzymes of the bee immune system that may contribute to oxidative stress (Sadd & Barribeau, 2013; Vanaerschot et al., 2014). For experiments, caffeic acid was dissolved to a concentration of 22.2 mM (4 mg mL−1) in 50% methanol and tested at final concentrations of up to 1.85 mM. This concentration is more than 30-fold the levels that occur in most types of honey (Can et al., 2015) and 10-fold the mean cinnamic acid concentration in pollen (Serra Bonvehi, Soliva Torrentó & Centelles Lorente, 2001). Therefore, the tested concentration range was likely to detect any direct effects against trypanosomatids that could be attributed to phytochemical consumption by bees.

The addition of sugar to the medium was also tested for positive or negative effects on growth. Bee-collected pollen, such as that used to create the extracts in this study, is rich in sugars from nectar, which are added to the pollen by bees during collection (Roulston & Cane, 2000). Previous experiments with C. bombi cell cultures showed that growth was strongly inhibited when 20% of growth medium was replaced by sugar syrup (Cisarovsky & Schmid-Hempel, 2014). However, many trypanosomatids prefer sugars to proline as a carbon source (Lamour et al., 2005; Bringaud, Rivière & Coustou, 2006), which suggests that addition of sugar to the tryptose- and liver-based growth medium (Salathé et al., 2012) could enhance growth. To evaluate the effects of pure sugars relative to pollen extracts, a 1.1 M sugar solution, consisting of equimolar amounts of glucose and fructose in 50% methanol, was added to the growth medium at final concentrations of up to 93 mM. This sugar concentration was chosen to slightly exceed the likely sugar concentration in the pollen extracts, which was estimated a priori as ∼500–660 mM. This estimate was based on a sugar content of ∼1.7–2.2 mol kg−1 (∼30–40% monosaccharides by weight in the pollen (Todd & Bretherick, 1942; Herbert & Shimanuki, 1978; Roulston & Cane, 2000; Campos et al., 2008)), with ∼30% pollen in the extract. The sugar composition was chosen to reflect the roughly equal amounts of glucose and fructose that have been found in nectar and honey (London-Shafir, Shafir & Eisikowitch, 2003; Ohmenhaeuser et al., 2013). Although pollen can contain sugars other than monosaccharides (Herbert & Shimanuki, 1978), monosaccharides were used because the bee intestine rapidly hydrolyzes disaccharides to glucose and fructose (Nicolson, 1998), which likely leaves only monosaccharides in the distal intestine where trypanosomatids become established (Lipa & Triggiani, 1988; Schwarz et al., 2015). The 1.1 M sugar solution was tested at up to 8.3% concentration by volume (93 mM monosaccharides in growth medium).

Experimental design

Each experiment tested the effects of treatments on growth of parasite cell cultures in 96-well microplates. The first experiment tested extracts of six different species of pollen. The second experiment tested the effects of buckwheat, rapeseed, and sunflower pollen extracts, individually and in a mixture that consisted of equal proportions of each of the three extracts (i.e., one-third buckwheat, one-third rapeseed, and one-third sunflower extract by volume). The third experiment tested the effects of added chemicals, which included the common floral phytochemical caffeic acid and a sugar solution. This third experiment included buckwheat pollen extract as a positive control to verify the effects of pollen extracts observed in the previous two experiments.

To test the effects of pollen extracts, extract of each of the six pollens was dissolved at six concentrations by two-fold serial dilution. Concentrations ranged from 0–5% (for pollen extracts) or 0–8.3% (for chemical additions) final concentration by volume in growth medium. Additional 50% methanol was added to samples of lesser concentrations to equalize methanol concentrations across samples. The inner 48 wells of a 96-well plate were filled with 100 µL of treatment solution (at 2× final concentration) and 100 µL of a suspension of C. bombi cells (103 cells µL−1), to achieve an initial cell density of 500 cells µL−1 (250 cells µL−1 for chemical addition experiment). Outer wells were filled with 200 µL sterile water to mitigate edge effects. Plates were sealed with laboratory film and placed inside zippered sandwich bags to minimize evaporation. Samples were incubated without shaking at 27 °C in a dark incubator. Growth was measured as optical density (630 nm) at 24 h intervals for 5 d by a spectrophotometer. Before each growth measurement, plates were shaken on a microplate shaker (30 s, 1,000 rpm, 4 mm orbit) to homogenize and resuspend the cells. In addition, immediately before each spectrophotometer reading, the plate cover used during incubation was exchanged for a dry plate cover under sterile conditions, to prevent condensation from interfering with optical density measurements. Wells that contained treatment media without cells and were used to control for changes in optical density independent of parasite growth. Experiments included n = 8 (for pollen extracts) or n = 5 (for chemical additions) replicate samples per treatment concentration, plus the n = 2 negative control wells of cell-free treatment medium. Final concentrations of methanol were 2.5% (for pollen extracts) or 4.17% (for chemical additions) by volume.

Analysis

Because no extracts or chemicals fully inhibited growth, EC50 values could not be calculated. Instead, we used linear regression to test for concentration-dependent changes in parasite growth. Treatment concentration (in percent extract for pollen extracts and mM for chemical additions) was used as the predictor variable. For the Caffeic acid + Sugar treatment in the chemical additions experiment, mM sugar was used as the predictor, to better compare the effects of sugar with versus without caffeic acid. Maximum optical density at 630 nm, estimated using a model-free spline fit, (Kahm et al., 2010), was used as the response variable. Separate models were fitted for each pollen extract or chemical. P-values were adjusted with a Bonferroni correction to account for multiple tests within each experiment. Graphs were produced with the R package ggplot2 (Wickham, 2009).

Chemical analyses

Sugar contents of each type of pollen extract were determined by HPLC (Alliance e2695 HPLC system, Waters Co., USA) coupled with an evaporative light scattering detector (ELSD, Waters 2424; Waters, Milford, MA, USA). Extracts were separated on a COSMOSIL Sugar-D column (4.6 mm I.D.  × 250 mm length; Nacalai Tesque, Kyoto, Japan) at a column temperature of 30 °C. The mobile phase was 80% acetonitrile and 20% methanol throughout the run, with a flow rate of 0.9 mL min−1. Quantities of sugar were determined based on external standard curves from analysis of pure solutions of aqueous fructose, glucose, and sucrose. Each sample had a total run time of 30 min (including column regeneration). Sugar contents are expressed as means of two (fructose and sucrose) or three (glucose) technical replicates.

Results

Extracts of each of the six pollens increased C. bombi growth, as measured by the maximum cell density achieved during the 5 d incubation period (Fig. 1); the increase in growth was significant in analysis by linear regression (Table 1). Relative to the pollen-free control, addition of 5% extract of each pollen resulted in approximately 50% higher maximum density. A mixture of buckwheat, rape, and sunflower extracts had effects that were similar to those of the individual pollens in isolation (Fig. 2).

Figure 1 Individual pollen extracts increased parasite growth.

Extracts of six types of pollen were tested at up to 5% concentration. Each panel (A–F) shows the maximum optical density (OD 630 nm) for parasites exposed to 50% methanol extracts of one of the six types of pollen. An additional 50% methanol was added to samples of lesser concentrations to equalize methanol concentrations (2.5% by volume) across samples. Points and error bars show means and standard errors for each concentration (n = 8). OD, optical density (630 nm). See Figs. S1–S3 for complete growth curves.

Table 1 Effects of pollen extracts and supplemental chemicals on growth.

Estimates and p-values are for linear regression after Bonferroni correction for multiple testing within each experiment. Coefficients are expressed as change in maximum optical density per percent pollen extract (A, B, and C: Buckwheat) or per mM chemical (C: Caffeic acid, Sugar, and Caffeic acid + Sugar).

Treatment	Coefficient (β)	Std. Error	T	p (T)	
A. Single pollens	
Buckwheat	0.037	0.0028	13.02	<0.001	
Lotus	0.024	0.0021	11.21	<0.001	
Poppy	0.034	0.0042	8.18	<0.001	
Rape	0.032	0.0028	11.25	<0.001	
Sunflower	0.021	0.0024	8.80	<0.001	
Tea	0.032	0.0031	10.28	<0.001	
B. Mixed Pollens	
Buckwheat	0.025	0.0021	11.99	<0.001	
Rape	0.036	0.0028	13.16	<0.001	
Sunflower	0.033	0.0025	13.36	<0.001	
Mixa	0.025	0.0016	15.08	<0.001	
C. Chemical additions	
Buckwheat	0.029	0.0030	9.95	<0.001	
Caffeic acid	−0.013	0.0068	−1.99	0.23	
Sugar	0.0034	0.00024	13.84	<0.001	
Caffeic acid + Sugarb	0.0028	0.00060	4.73	<0.001	
Notes.

a Mix treatment consisted of equal proportions of buckwheat, rape, and sunflower extracts.

b Coefficient expressed as change in OD per mM sugar.

Figure 2 Mixed and individual pollen extracts each increased growth.

The treatments consisted of extracts of (A–C) individual pollens and (D) a mixture (“mix”) of equal proportions of buckwheat, rapeseed, and sunflower pollen. Each panel represents a different pollen extract. An additional 50% methanol was added to samples of lesser concentrations to equalize methanol concentrations (2.5% by volume) across samples. Points and error bars show means and standard errors for each concentration (n = 8).

In HPLC analyses, pollen extracts were found to contain considerable amounts of sugar, mainly the monosaccharides fructose and glucose (Fig. 3), with sucrose found in tea and sunflower extracts, but in relatively small amounts. Total sugar content ranged from 192 mM in tea pollen extract to 789 mM in rape pollen extract.

Figure 3 Sugar composition of pollen extracts.

Gray bars represent fructose; orange bars represent glucose; blue bars represent sucrose. Concentrations were determined by HPLC with refractive index detector. Bars show mean of technical replicates (2 for fructose and sucrose, 3 for glucose).

In the test of specific additional chemicals, both the buckwheat pollen extract (positive control) and addition of sugar solution resulted in increased growth (Fig. 4 and Table 1), whereas the common floral phytochemical caffeic acid had only weakly inhibitory effects at up to 1.85 mM (Fig. 4 and Table 1), which is an order of magnitude higher than any concentration documented among different types of honey (Can et al., 2015). Additional sugar had similar effects whether it included caffeic acid (coefficient = 0.0028 ± 0.00060 SE) or not (coefficient = 0.0034 ± 0.00024 SE, Table 1). Although a transient decrease in growth was found at intermediate caffeic acid concentrations (Fig. S3), the greatest growth inhibition occurred at intermediate concentrations, suggesting that inhibition reflected the position of the sample on the plate rather than the effects of the phytochemical. As can be seen from the growth curves of the buckwheat, sugar, and caffeic acid + sugar treatments, early growth was often poor in the samples of intermediate concentration that were incubated in the center of the plate. We attribute this effect to toxicity of the methanol, which would have dissipated relatively slowly from the samples in the center of the plate as compared to the samples at the periphery. In previous tests of three different C. bombi strains, including the IL13.2 strain used here (Palmer-Young et al., 2016), only weak effects of caffeic acid occurred at concentrations up to 1.39 mM (Fig. S4). None of the tested concentrations resulted in >50% growth inhibition, which precluded estimation of an EC50 concentration.

Figure 4 A floral phytochemical had weak effects on growth, whereas supplemental sugar increased growth.

Each panel shows the growth curve for parasites exposed to one of the chemical treatments. (A) Buckwheat pollen extract was used as a positive control to confirm increased growth in the presence of pollen extract. (B) The sugar treatment consisted of equimolar amounts of glucose and fructose; both (B) caffeic acids and (C) sugars were dissolved in 50% methanol. (D) In the caffeic acid + sugar treatment, concentrations are shown for caffeic acid (top line) and sugars (bottom line). Additional 50% methanol was added to samples of lesser chemical or extract concentrations to equalize methanol concentrations (4.17% by volume) across samples. Points and error bars show means and standard errors for each concentration (n = 5).

Discussion

These experiments indicate that pollen extracts can increase growth of an intestinal parasite, and that the growth-promoting effects of pollen extracts can be reproduced by addition of similar amounts of a sugar solution. Pollen phytochemicals appear to be insufficient to stop growth of C. bombi, and moreover, pollen appears to contain substances that improve trypanosomatid growth. This result is consistent with previous experiments that showed a decrease in infection intensity in bees deprived of pollen; our results suggest a mechanism by which pollen may directly promote trypanosomatid infection. The positive effects of pollen nutrients on C. bombi, a hindgut trypanosomatid, suggests the potential for facilitation of nutrient-limited hindgut parasites by midgut parasites that interfere with nutrient absorption. In addition, the high phytochemical tolerance of C. bombi relative to that of bloodstream trypanosomatids invites further study on adaptation to phytochemicals in different trypanosomatid species, and variation in tolerance across life stages.

Phytochemical insensitivity

Crithidia bombi growth was not inhibited by any of the pollen extracts. This was unexpected in the context of current literature on bumble bee-Crithidia interactions, which has suggested that phytochemical ingestion can reduce C. bombi infection (Manson, Otterstatter & Thomson, 2010; Baracchi, Brown & Chittka, 2015; Richardson et al., 2015). On the contrary, growth was increased by addition of pollen extracts. Similarly, growth of C. bombi in the present study was only weakly inhibited (<50% decrease in maximum OD, Fig. 3) by caffeic acid at concentrations of over 1.8 mM. From an ecological perspective, the 1.8 mM concentration is 6-fold greater than the concentration found in pollen (0.3 mM (Šarić et al., 2009)) and 12-fold greater than the 0.15 mM (26.8 ppm) maximum concentration found in any type of honey, including the Quercus spp. honey from bees that foraged on oak tree sap (Can et al., 2015). No other type of honey exceeded 0.05 mM (8.8 ppm (Can et al., 2015)). Hence, it appears that C. bombi is little inhibited by naturally occurring concentrations of caffeic acid or other compounds that may have been present in the six different pollen extracts. The insensitivity of C. bombi to naturally occurring levels of hydroxycinnamic acids is consistent with previous results (Palmer-Young et al., 2016).

The phytochemical tolerance observed here is greater than that found in some bloodstream-form trypanosomatids, but not unprecedented for trypanosomatid life stages found in insects. Concentrations needed for 50% growth inhibition (EC50) ranged from 0.006 to 0.31 mM (1.1–56 ppm) caffeic acid in bloodstream forms of L. donovani, T. brucei, and T. cruzi, (Tasdemir et al., 2006; Grecco et al., 2014). However, tolerance can be much higher in other trypanosomatid species and life stages. Leishmania promastigotes, the stage found in the insect gut, retained viability at concentrations comparable to those tested in our study, with inhibitory concentrations of 1.05 mM for Leishmania amazonensis and >2.78 mM for L. braziliensis (Passero et al., 2011). Similarly, caffeic acid EC50 was only 3.9–6.1 nM for intracellular bloodstream amastigotes of four tested Leishmania spp., but >2800 nM (2.8 µM) for promastigotes, the stage found in the insect gut; the same trend was observed for seven other compounds (Radtke et al., 2003). The higher phytochemical tolerance of extracellular, promastigote Leishmania relative to bloodstream-form, intracellular amastigotes was confirmed in other studies (Kolodziej & Kiderlen, 2005). These differences in sensitivity may reflect different levels of exposure to phytochemicals during different life stages, or costs of resistance in bloodstream forms. More study is needed to understand the basis of differential resistance across life stages and species, which could be relevant to development of drug resistance.

Commensurate with its evolutionary history of chronic phytochemical exposure in the bee gut, C. bombi appears to be well adapted to phytochemicals, including those that are toxic to other trypanosomatids and even those initially shown to reduce infection intensity. For example, C. bombi exhibited EC50 values for several phenolics that were orders of magnitude higher than those reported for other trypanosomatid species (Palmer-Young et al., 2016), and infection was robust to thymol, anabasine, and nicotine under controlled conditions (Biller et al., 2015; Thorburn et al., 2015). Although the present study did not address possible host-mediated effects of phytochemicals on infection, such as phytochemical-induced stimulation of immune responses (Borchers et al., 1997; Mao, Schuler & Berenbaum, 2013) or changes in gut kinetics (Tadmor-Melamed et al., 2004) that have been observed in other species and could alter trypanosomatid attachment to the gut wall (Schwarz et al., 2015), the fact that none of the six pollens inhibited growth demonstrates that C. bombi is robust to many of the phytochemicals in the diet of its hosts.

Many trypanosomatids complete their life cycle in two hosts, which may include insects, mammals, and plants (Maslov et al., 2013). It would be intriguing to use the comparative method to test whether evolutionary history is predictive of phytochemical tolerance. To accomplish this, future studies could compare phytochemical tolerance of species that occupy niches with different levels of phytochemical exposure. In order of decreasing intensity of phytochemical exposure, these could include (a) species that utilize plants as hosts, (b) those that are gut parasites of herbivores, and (c) species and life stages that live in the blood and are transmitted by blood-feeding insects. Trypanosomatids with an evolutionary history of phytochemical exposure would be expected to have higher phytochemical tolerance than those that encounter phytochemicals only occasionally or indirectly.

Increase in growth was reproduced by addition of sugar

The growth-promoting properties of the pollen extracts may be attributable to their constituent nutrients, in particular monosaccharides, and possibly amino acids. Pollen collected by corbiculate bees, such as bumble bees, is moistened with nectar, which renders it sufficiently sticky to be carried in the bee’s corbicula (pollen basket) (Roulston & Cane, 2000). Thus, bee-collected pollen contains considerable amounts of carbohydrate, including up to 40% sugars by weight (Todd & Bretherick, 1942; Roulston & Cane, 2000). Although the osmolarity of very high (∼20% w/v, unspecified composition) sugar concentrations can kill C. bombi as well as other microbes (Cisarovsky & Schmid-Hempel, 2014), the effects found in this study indicate that addition of sugar at low concentrations is beneficial for trypanosomatid growth.

The monosaccharides added to the growth medium would have increased the sugar content of growth medium several-fold, providing up to 93 mM monosaccharides in addition to the 12.2 mM in the base medium (10 mM from fructose + 2.2 mM from glucose in liver broth (Salathé et al., 2012)). This additional sugar may have increased the quality of the media for C. bombi energy production. Insect and bloodstream-form trypanosomatids can use glucose as a carbon source (Mazet et al., 2013). Although proline is the normal carbon source for trypanosomatids in insect guts where glucose is scarce (Bringaud, Rivière & Coustou, 2006), proline metabolism is dramatically reduced in the presence of glucose, which suggests that glucose is a preferred energy source (Lamour et al., 2005). The trypanosomatids that infect bees, which consume carbohydrate-rich diets, may be particularly adapted to use of carbohydrates. In a genomic comparison between Leishmania major and the honey bee gut parasite Lotmaria passim (n.b. Originally reported as C. mellificae), genes related to carbohydrate metabolism were enriched in the bee parasite compared to its bloodborne relative (Runckel, DeRisi & Flenniken, 2014). Genomic studies may reveal whether carbohydrate metabolism is also well developed in C. bombi, which is a close relative of L. passim (Schwarz et al., 2015).

The stimulatory effect of sugar on C. bombi growth raises the question of possible facilitation of hindgut trypanosomatid infections by co-occurring infections that impair nutrient absorption, such as Nosema ceranae and Nosema apis. Nosema spp. have been implicated in collapse of honey bee colonies (Higes et al., 2009; Cornman et al., 2012) and may infect bumble bees as well (Graystock et al., 2013). Field studies have found positive correlations between Nosema apis and trypanosomatid infections in honey bees (Cornman et al., 2012), which provides suggestive evidence for positive effects of Nosema spp. on gut trypanosomatids. The present study suggests a mechanism by which Nosema infection could contribute to trypanosomatid infection via negative effects on sugar absorption in bees. Healthy bees and other nectivorous insects have an excellent ability to digest and absorb sugars from nectar (Nicolson, 1998), which may explain why there was no effect of dietary sugar concentration on infection intensity in B. impatiens (Conroy et al., 2016) . However, gut infection by microsporidians can disrupt the midgut epithelium (Higes et al., 2008). Injury to midgut tissue may decrease absorption of sugar in hosts, as suggested by the decreased hemolymph sugar concentrations and increased hunger observed in Nosema-infected bees (Mayack & Naug, 2009). As a result of Nosema induced malabsorption, more sugar may reach the hindgut, and thereby increase the supply growth-limiting carbohydrate to trypanosomatids. To test whether Nosema related malabsorption facilitates infection by trypanosomatids, future experiments could test the effects of microsporidian infection on fecal carbohydrate content and trypanosomatid infection intensity.

Although pollens differed considerably in sugar content, each extract had similar effects on growth. This suggests that, in addition to sugars, pollen may contain additional substances that promote C. bombi growth. One such substance may be proline, which is the normal carbon source for trypanosomatids in insect guts where glucose is scarce (Bringaud, Rivière & Coustou, 2006). Proline has many functions in plants, including regulation of osmolarity and resistance to abiotic stress (Verbruggen & Hermans, 2008), and is the most abundant amino acid in pollen. For example, pollen of nine Asteraceae species contained 85–420 mmol kg−1 free proline (Mondal, Parui & Mandal, 1998); another study found 173 mM kg-1 free proline in Spanish bee pollens (Paramás et al., 2006). A proline content of 200 mmol kg−1 in pollen would correspond to approximately 60 mM proline in the pollen extract and 3 mM additional proline in samples treated with 5% pollen extract. This would roughly double the 2.5 mM proline in the base growth medium (Salathé et al., 2012). Although we were not able to conduct an amino acid analysis, which requires a special type of HPLC column, proline is generally agreed to be found in all pollens (De Simone et al., 1980) as the dominant amino acid, composing 15–69% of total amino acids (Yang et al., 2013). More study is needed to determine which, if any, additional pollen compounds alter trypanosomatid growth in vitro and in vivo.

Pollen in pollinator communities

Pollen, like nectar, is an indispensable source of nutrients for bees and other pollinators that supports insect immunity (Brunner, Schmid-Hempel & Barribeau, 2014), survival (Conroy et al., 2016), and reproduction (Vaudo et al., 2015). However, pollen may nourish parasites as well as hosts. Although phytochemicals may reduce growth of some microbes, parasites of phytophagous animals are likely adapted to the phytochemicals and concentrations found in the diets of their hosts. Moreover, in coevolved obligate parasites such as Crithidia and other trypanosomatids (Maslov et al., 2013), there may be substantial overlap in the nutrient requirements of parasites and hosts. Shared nutritional requirements, and the increased fitness of parasites in well-nourished hosts, may result in tradeoffs between starvation of parasites and starvation of hosts, and may explain the utility of anorexia as a defense against parasites in some taxa (Parker et al., 2011). The results found here exemplify the trade-offs between host health and defense, underline the difficulty of eradicating well-adapted parasites without compromising host fitness, and suggest that natural selection may act across all levels of tritrophic interactions.

Supplemental Information

Supplemental Information 1 Supplementary materials: Data files for effects of pollen on C. bombi

Dataset 1: Effects of extracts of individual pollens.

Dataset 2: Effects of individual and mixed pollen extracts.

Dataset 3: Effects of chemical additions.

“Row” and “column” refer to locations on the 96-well test plate

“conc” indicates the concentration in percent extract (or stock solution) by volume in the final sample

“net0” through “net24” are the net OD readings (difference in OD 630 nm between samples and cell-free controls at the corresponding concentration and timepoint) at each timepoint, with time given in hours.

“integral” is the estimated area under the curve of OD vs time, calculated by the “grofit” function in R

Dataset 4: Pollen extract sugar contents. Contents of fructose, glucose, and sucrose are given in mg per mL and mM

Click here for additional data file.

Supplemental Information 2 Supplementary Figures for Palmer-Young EC, Pollen extracts increase growth of a trypanosome parasite of bumble bees

Supplementary Figure 1. Growth curves for the individual pollen experiment.

Supplementary Figure 2. Growth curves for the mixed-pollen experiment

Supplementary figure 3. Growth curves for the chemical additions experiment.

Supplementary figure 4. Effects of caffeic acid (aqueous) on three strains of Crithidia bombi.

Click here for additional data file.

Thanks to Lynn Adler, Jonathan Giacomini, and Rebecca Irwin for inspiring the study; to Ben Sadd for providing the parasite cell culture; to Rob Wick, William Manning, and Jeffrey Blanchard for sharing laboratory space and equipment; to Anastasiya Mirzayeva for technical assistance; to Matthias Kahm, Hadley Wickham, and the R Core team for statistical resources; and to the reviewers (Frederic Bringaud and Julius Lukes) and editor Christine Clayton for their critique and improvement of the manuscript. Additional thanks to Phil Stevenson for coordinating analysis of pollen extracts.

Additional Information and Declarations

Competing Interests

Author Contributions

Data Availability

The authors declare there are no competing interests.

Evan C. Palmer-Young conceived and designed the experiments, performed the experiments, analyzed the data, contributed reagents/materials/analysis tools, wrote the paper, prepared figures and/or tables, reviewed drafts of the paper.

Lucy Thursfield conceived and designed the experiments, performed the experiments, analyzed the data, contributed reagents/materials/analysis tools, wrote the paper, reviewed drafts of the paper.

The following information was supplied regarding data availability:

The raw data has been supplied as a Supplementary File.

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
