# Peer review of "Pollen extracts and constituent sugars increase growth of a trypanosomatid parasite of bumble bees"

_PeerJ, doi:10.7717/peerj.3297_

## Round 0.1 · original submission · Major Revisions

Please in particular address the concerns of reviewer 1, in detail. Provide data concerning the doubling time during exponential growth, measured by diluting the cells as necessary to maintain growth, with and without the extracts. Independently, measure the maximal density. Determine the EC50. Give all concentrations in mM or micromolar, as is standard scientific practice. Ppm is not acceptable.

In addition, the authors must correct all incorrect nomenclature, including the incorrect designation of Crithidia as a "trypanosome" in the title. Please read (and cite) the literature suggested. The part about T. brucei appears to be irrelevant and should be removed.

·

Basic reporting

no comment

Experimental design

1- All the data presented in this manuscript are based on growth analyses of Crithidia bombi, a parasite found in the gut of bumble bees, incubated in vitro in the presence of pollen extracts or pollen components, such as sugars and caffeic acid. To determine the "growth efficiency" the authors measured OD630 after five days of growth with an initial density of 5x10e5 cell per ml. According to the supplementary informations (Figure S1), in this experimental set up, cells incubated in standard medium grow exponentially during the first 2 days, growth stop at the end of the third days and then cell density falls down during the next 2 days, certainly because of cell death. Then during the incubation time (5 days) the OD is a combination of exponential growth and cell death, which is not appropriate to determine the effect of pollen extracts or pollen components on growth.
To address this question two kinds of complementary experiments could be performed, (1) determining the maximum growth density and (2) determination of the doubling time by maintaining the cell in exponential phase. Clearly the doubling time is not affected by pollen extracts since the growth rate is the same for all conditions during the first two days (Figure S1). However, the maximum growth density is considerably increased in the presence of pollen extracts at day 3 (or day 4). All along the analysis, the author should consider the maximum growth density, instead of the density at day 5.

3- In Table 1, "slope" should be replaced by maximum cell density and the author should explain how the values have been calculated.

4- All along the manuscript sugar concentration are expressed in % (probably w/v) and in Figure 3, concentration in abscissa is expressed as percent by volume. To help the readers all concentrations should be express in mM and not in %, including n the abscissa of Figure 3. For instance, the calculation in sentence the "the sugar solution was tested at up to 8.3% concentration by volume (16.7 g/L monosaccharides in growth medium)" (lanes 166-167) is strange. Indeed, 2 x 8,3% = 16.6% which should mean 16.6 g/100 ml not 16.7g/L. All these concentration values should be double checked and clarified.

5- Lines 108-113, it would be relevant to describe the growth medium and include the sugar concentration (in mM).

6- Line 143 (and in the whole manuscript). Concentrations should be expressed in mM instead in ppm.

7- Line 180, what does mean "0-8.3% final concentration by volume in growth medium" (for chemical. Again mM should be used.

8- Since sugars present in the pollen are responsible for the increase maximal growth of Crithidia, it would be relevant to determine the sugar concentration in pollen extracts (in mM of course).

9- The final concentration of methanol in each well should be clearly mentioned in the Materials and Methods. This section is not clear. It seems that the final concentration is 2.5% (v/v).

Validity of the findings

A number of interpretations of the data need to be reconsidered.

1- The increase of maximum cell density is certainly due to the increase of sugar concentration in the medium as proposed in the discussion (lane 279). The medium contains 2.2 g/l of hexose (11 mM), while pollen extracts provide 8-fold more sugars (16.7 g/l, 84 mM). This is the most important data of the manuscript, which should appear in the result section as well as in the title, such as for instance " Sugars in pollen extracts increase growth of a trypanosome parasite of bumble bees".

2- Figure S3, used to calculate numbers presented in Table 1, clear shows that caffeic acid affects the doubling time, since a significant growth retardation is observed during the first two days of growth. Unfortunately, the author only considered the OD after 5 days of growth, which is not strongly affected, to state that caffeic acid has "weak inhibitor effects on Crithidia bombi". This is a clear misinterpretation of the data and the authors should compare the doubling time of the parasite in the presence and the absence of caffeic acid, and modify all the sentences describing the effect of caffeic acid.
However, it seems that the presence of sugar inhibits the effect of caffeic acid (Figure S3). Again, this experiment should be done properly by determining the doubling time of the parasite with the different combinations of compounds.
In conclusion of this aspect, the experiments has to be done properly by measuring doubling time, or better by determining the EC50 as mentioned in lane 243, and the text should be changed accordingly. For instance in line 233, one can read "Crithidia bombi growth was not inhibited by any of the pollen extracts", which is wrong according to Figure S3.

Additional comments

1- All along the manuscript, the authors have to use "trypanosomatid", "Trypanosomatidae" or "Crithidia" instead of "trypanosome" to qualify Crithidia bombi. Trypanosoma (Trypanosomes) and Crithidia are distinct genera of the Trypanosomatidae (trypanosmatid) family, which is composed of 15 genera. Actually, Trypanosoma and Crithidia belong to two different subfamilies. So clearly Crithidia species are clearly not trypanosomes.

2- Similarly the word "antitrypanosomal" is not appropriate.

3- Line 90, increased should not be in italic.

4- Line 230, explain what are "bloodstream trypanosomes".

5- In lines 281-285, the author compare the procyclic form of Trypanosoma brucei, which is present in the insect vector, called trypanosomes with Crithidia also called trypanosomes. Both parasites should be clearly differentiated in te text. The sentence "trypanosomes rely on glycolysis for energy production (Mazet et al.2013)", only concerns the mammalian form of T. brucei, not the insect form, and is not relevant for the discussion. The procyclic form of T. brucei, don't use peptides as carbon sources, but proline. Please, correct this part of the discussion by using the correct words and references.

·

Basic reporting

This is an interesting paper, as it studies the relationship between trypanosomatids and their insect hosts from a novel perspective. The finding that - against expectations - pollen extracts stimulate growth of these flagellates is worth publishing.
Moreover, it is very relevant because it brings the study of increasingly endangered bumble bees and their highly prevalent parasites/commensals on another level. I would very much appreciate a follow-up (indicated by the author), in which the relationship between trypanosomatids, Nosema and their host would be analyzed.

Experimental design

I have no problem with the experimental design, it is simple yet well thought thru.

Validity of the findings

The findings and conclusions are supported by the data.

Additional comments

I have a few comments, which may be considered by the author.
First of all, the author considers trypanosomatids in every context as parasites. I think that's not the case and there is some literature indicating that they may behave as commensals. This should be added at least as a note. There is an important paper in this context, by Hamilton et al., mBio 2015, that was apparently overlooked.
The author confuses the terms trypanosomes and trypanosomatids, so please distinguish between them carefully. Possible inclusion of a sentence or two on the phylogenetic placement of the studied flagellate would be beneficial for the reader.

PS: remove italics, line 90

---

## Round 0.2 · accepted · Accept

Both referees were very happy with the improvements that you made to your paper.

·

Basic reporting

The authors appropriately answered the reviewer's comments

Experimental design

The authors appropriately answered the reviewer's comments

Validity of the findings

The authors appropriately answered the reviewer's comments

Additional comments

The authors appropriately answered the reviewer's comments

·

Basic reporting

I am happy with the resubmission.

Experimental design

I am happy with the resubmission.

Validity of the findings

I am happy with the resubmission.

Additional comments

The paper has been significantly improved and is in my opinion suitable for publication.